# Total water level driving processes influence the potential for coastal change along United States coastlines

Gabrielle P. Quadrado[1,2] ⓘ and Katherine A. Serafin[1] ⓘ

[1]Geography, University of Florida, USA and [2]College of Engineering & Computer Science, University of Central Florida, USA

## Research Article

total water level; wave runup; storm impact regimes; coastal hazards

**Corresponding author:**
Gabrielle P. Quadrado;
Email: gquadrado@ucf.edu

## Abstract

Effectively managing coastal impacts requires understanding how hydrodynamic processes interact with beach morphology and alter sandy beaches. Sallenger's Storm Impact Regime classification provides proxies for identifying these interactions by classifying potential coastal changes into impact regimes based on the elevation of total water levels (TWLs), the combination of waves, tides and nontidal residuals, compared to the elevation of beach morphology features. Here, we evaluate spatiotemporal variations in TWL drivers during storm impact regimes across sandy beaches along the continental U.S. coastlines. TWL magnitude and composition vary across impact regimes and regions. Although impacts are identified using consistent definitions, the processes contributing to regimes are location-specific, influencing where and how often impacts occur. Wave runup is the dominant contributor to TWLs in all regimes, but its contribution is gradually offset by increases in the nontidal residual as storm impacts intensify, despite waves becoming more energetic during storms. The Pacific and Atlantic coasts show the highest susceptibility to coastal impacts due to high average TWLs relative to morphological thresholds. Our findings identify regional differences in TWLs with potential for coastal impacts, offering critical insights into how large-scale changes in individual processes may influence local coastal hazards along open sandy coastlines.

## Impact statement

This research provides key insights into the drivers of potential impacts at sandy beaches along the continental United States coastlines, revealing variations in how waves, tides and storm surge combine to cause local coastal hazards, such as dune erosion and flooding. While coastal hazards are consistently identified by water level exceedance over physical thresholds, the magnitude and combination of ocean processes that lead to flooding and erosion vary by location. Consequently, the likelihood and frequency of coastal impacts are site-specific. Our findings show that waves are generally the main driver of coastal impacts, but their influence is offset by the increasing importance of storm surge, tides and other factors, such as sea level anomalies and seasonal sea level changes, as impacts become more severe. Results also indicate a growing influence of storm surge as impacts intensify, with its contribution peaking during flooding. Understanding how different ocean processes combine when driving coastal impacts helps planners, engineers and emergency managers identify which local factors should be prioritized when projecting current and future risks to coastal hazards. Although this study focuses on the United States coastlines, the approach we present is broadly applicable to assess coastal hazards across other coastal communities worldwide. By identifying the drivers of potential impacts in different locations, our work can support the development of more effective risk assessments, early warning systems and adaptation strategies in the context of a changing climate.





## Introduction

Nearly a quarter of the world's sandy beaches are eroding at rates exceeding 0.5 m/yr, with about one in six beaches eroding at rates exceeding 1 m/yr (Luijendijk et al., 2018). This widespread erosion, often linked to storm-induced water levels, increases the exposure of coastal communities worldwide to flooding and erosion hazards (Neumann et al., 2015; Luijendijk et al., 2018). In the United States (U.S.), property loss due to coastal erosion costs ~$500 million per year (US Climate Resilience Toolkit, 2021), while a single hurricane-induced coastal flooding event can lead to billion-dollar damages (NOAA NCEI, 2023). In addition, ~85% of sandy shorelines along the U.S. Atlantic and Gulf coasts are eroding (Luijendijk et al., 2018; The Heinz Center, 2000), while ~36% are experiencing erosion on the Pacific Northwest (PNW) and the California coasts (Hapke et al., 2006; Ruggiero et al., 2013). As hazards occur widely along U.S. coastlines, it

becomes increasingly important to understand the combination of processes driving high-water-level events that result in coastal impacts.

Process-based models are widely used in coastal change assessments for their ability to predict detailed changes in morphological response to hydrodynamic forces like waves, tides, storm surge and sea level rise (e.g. Vousdoukas et al., 2011; Callaghan et al., 2013; Bilskie et al., 2016; Passeri et al., 2018; Splinter et al., 2018; Cohn et al., 2019; Sherwood et al., 2022). While capturing fine-scale interactions between coastal morphology and hydrodynamic processes, process-based models are usually applied to specific events at specific locations, as they become computationally expensive at hourly resolutions over spatial domains of hundreds of kilometers and temporal scales from months to decades. As coastal management and adaptation often require a comprehensive understanding of morphodynamics along kilometer-long coastlines, additional techniques are necessary to integrate detailed model insights across broader spatial scales.

Due to the challenges of modeling morphologic change over large spatiotemporal scales, coastal change and associated impacts have historically been represented by proxies, which infer beach morphology changes driven by hydrodynamic processes (e.g., Sallenger, 2000; Ruggiero et al., 2003; Stockdon et al., 2007; Aucelli et al., 2018; Leaman et al., 2021; Leung et al., 2024). Sallenger (2000) introduced the Storm Impact Scale to characterize storm impacts on barrier islands by comparing high total water levels (TWLs), the combination of waves and measured water levels at the coast, to thresholds based on the elevation of morphologic features, such as the dune crest or the dune toe. In this scale, four distinct storm impact regimes are defined: swash, collision, overwash/overtopping and inundation, which represent, respectively, beach face changes, base of the dune erosion, dune erosion and overtopping, and flooding (Sallenger, 2000). Stockdon et al. (2007) demonstrated the predictive skill of Sallenger's storm-impact scaling model for estimating coastal responses to storms, showing that spatial variability in estimated storm responses is driven by variations in dune crest and wave runup elevations. In contrast, increases in significant wave height and storm surge as storms move onshore explain the temporal variability in coastal responses. Due to its effectiveness, the Storm Impact Scale has been applied worldwide to characterize storm impacts on open coasts (e.g., Ciavola et al., 2011; Splinter et al., 2018; Serafin et al., 2019; Stockdon et al., 2023; Turner et al., 2024).

TWL magnitude, however, is not the only factor that determines coastal impacts; the TWL relative composition is also critical. For instance, on dissipative beaches, the same peak TWL can result in either dune accretion or erosion depending on the relative contributions of waves and the still water level (SWL) (Cohn et al., 2019), which comprises astronomical tide, storm surge and mean sea level, to the TWL. In locations where the SWL dominates the TWL magnitude, the overall TWL is less influenced by wave transformation across the continental shelf than in places where waves are the main TWL driver (Serafin et al., 2019). Lastly, Theuerkauf et al. (2014) indicated that sea level anomalies persisting longer than 2 weeks can aggravate coastal erosion even without a storm. Attributing high water levels to different physical processes thus improves our understanding of coastal impacts, and such insights may help predict local flooding and erosion.

Several studies have evaluated the relative composition of extreme TWLs. Wave runup, the elevation reached by water on the beach due to wave breaking, is the dominant contributor to extreme TWLs at the global scale (Vitousek et al., 2017), and regionally along the U.S. Pacific (Serafin et al., 2017) and Atlantic coasts (Quadrado and Serafin, 2024). Wave runup is also the main contributor to peak TWLs during tropical cyclones on the U.S. Atlantic coast (Hsu et al., 2023), and it delivers much of the energy responsible for dune and beach erosion (e.g., Ruggiero et al., 2001; Stockdon et al., 2007; Cohn et al., 2019, 2021). This makes wave runup a critical process for understanding the likelihood of coastal change (e.g., Sallenger, 2000; Stockdon et al., 2007, 2023), particularly when combined with SWLs. Although studies have explored the drivers of coastal hazards by analyzing TWL components, few have related them to beach elevation, which is an essential link for understanding coastal change and hazard potential.

Here, we evaluate spatiotemporal variations in the processes contributing to TWLs impacting the beach face and backshore features at sandy beaches along the U.S. coastlines from 1980 to 2021. We pair wave hindcast data to measured water levels at 26 locations along the US Pacific, Gulf and Atlantic coasts and generate hourly TWL time series. For each site, we determine whether hourly TWLs reach or exceed Light Detection and Ranging (LiDAR)-derived local morphological thresholds based on the vertical heights of backshore characteristics related to Storm Impact Scale regimes (Sallenger, 2000; Stockdon et al., 2007). Then, we compute the relative contribution of the processes driving TWLs over each threshold and compare results across sites. We test whether TWL magnitudes differ significantly across impact regimes at each location, and across locations within each regime, using one-way analyses of variance (ANOVAs). Finally, we assess the likelihood of coastal change over the past decades by assigning TWL percentiles to morphological thresholds associated with each storm impact regime. Ultimately, our findings reveal spatiotemporal patterns in coastal impact drivers along the US coastlines, providing critical insights into understanding local coastal hazards on sandy beaches.

## Methods and Datasets

TWLs are defined by adding the SWL to wave runup (R) (Equation 1).

$$TWL = SWL + R \tag{1}$$

SWL time series, referenced to the North American Vertical Datum of 1988 (NAVD88), are extracted from 26 National Oceanic and Atmospheric Administration (NOAA)-operated water level stations located near open-coast sandy beaches along the Pacific, Gulf and Atlantic coastlines of the United States (Figure 1). Stations are selected based on record length ($\geq$ 30 years), geographic location ($\geq$ 50 km apart) and completeness ($\geq$ 80% of hourly data between 1980 and 2021). At each station, we further evaluate hourly completeness each year, and exclude years with < 80% hourly data, since missing 20% or more (i.e., $\approx$ 73 days) can bias findings toward times of the year with available data and potentially exclude extreme events causing impacts. We decompose the SWL signal into long-term relative sea level ($\eta_{RSL}$), astronomical tide ($\eta_A$) and the nontidal residual ($\eta_{NTR}$), which is further separated into seasonality ($\eta_{SE}$; intra-annual signal), storm surge ($\eta_{SS}$; high-frequency variability due to wind setup and atmospheric pressure anomalies), sea level anomalies ($\eta_{SLA}$; interannual to interdecadal variability associated with weather and climate patterns, current positions and coastal-trapped waves) and a residual signal ($\eta_{residual}$; difference between the measured and reconstructed SWL), such that

$$SWL = \eta_{RSL} + \eta_A + \eta_{NTR} \tag{2}$$

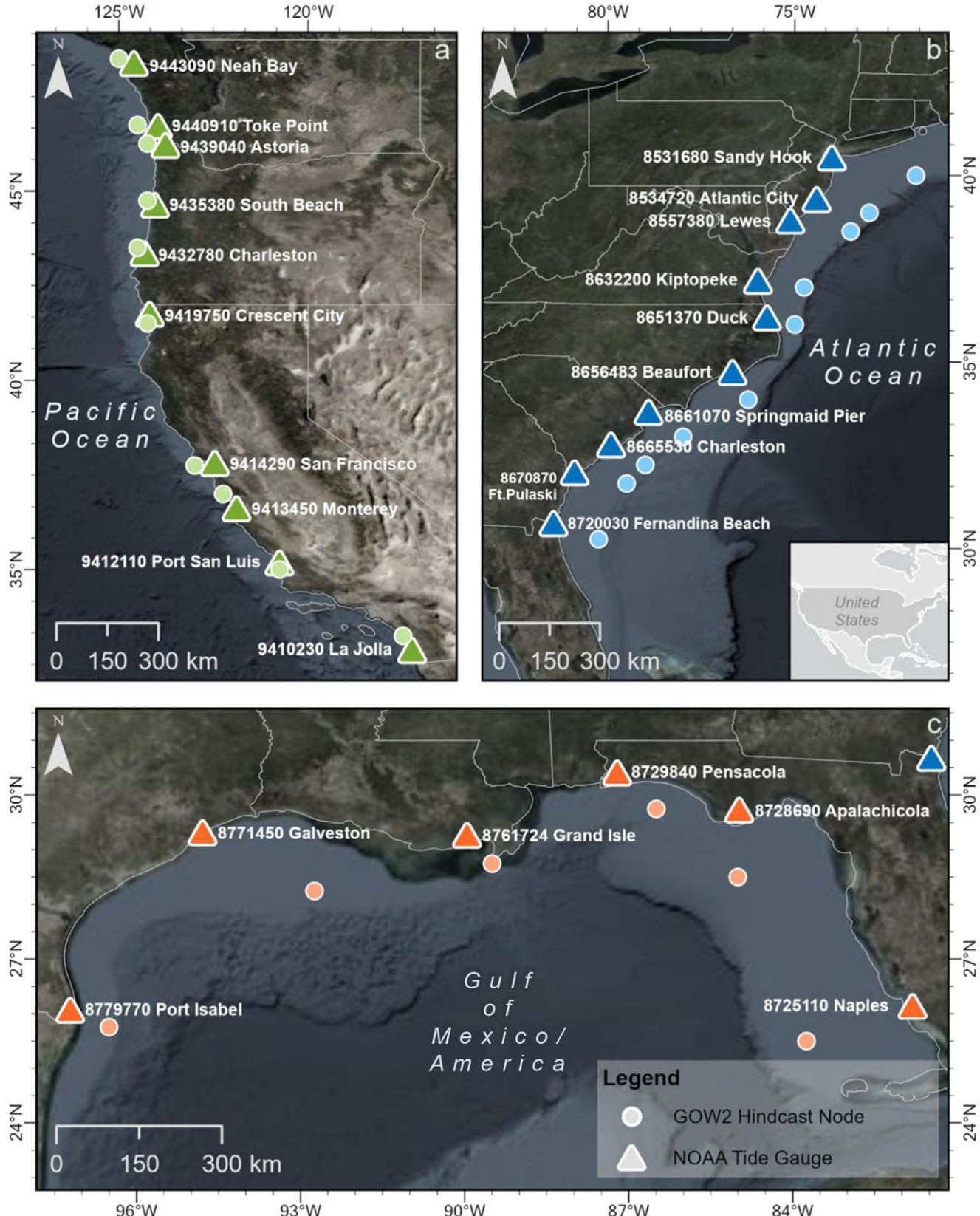

**Figure 1.** Map of locations of the water level stations (triangles) and wave hindcast nodes (circles). Colors represent the US coastlines: green for the Pacific (a), blue for the Atlantic (b) and orange for the Gulf (c).

$$\eta_{\text{NTR}} = \eta_{\text{SE}} + \eta_{\text{SLA}} + \eta_{\text{SS}} + \eta_{\text{residual}} \tag{3}$$

following the methods described in Quadrado and Serafin (2024) and briefly summarized in the Supplementary Material (Supplementary Text S1).

Wave runup is defined as the 2% exceedance value of wave runup ($R_{2\%}$) and is estimated using the Stockdon et al. (2006) formulation

$$R_{2\%} = 1.1 \left( 0.35\beta (H_0 L_0)^{\frac{1}{2}} + \frac{\left[ H_0 L_0 \left( 0.563\beta^2 + 0.004 \right) \right]^{\frac{1}{2}}}{2} \right) \tag{4}$$

where $\beta$ is the beach slope, $H_0$ is the deep-water significant wave height and $L_0$ is the deep-water wavelength, a function of the wave period. We select the Stockdon et al. (2006) equation for its predictive skill and validated performance across field and experimental studies (e.g., Stockdon et al., 2007, 2014; Matias et al., 2012; Plant and Stockdon, 2015; Atkinson et al., 2017; Kim et al., 2023). The empirical equation was developed with observations from various sandy beaches, including locations at or close to our study sites (i.e., Duck, NC; La Jolla, CA; and Agate Beach, OR), making it effective across different morphodynamic states and wave conditions (Kim et al., 2023).

To ensure wave forcings are from consistent depths, completeness, spatial resolution, and similar in record length, significant wave height and peak wave period data are obtained from the Global Ocean Waves version 2.0 hindcast (Pérez et al., 2017). We colocate grid cells with the NOAA water level data at ~100 m contour, which is the closest depth to shore consistently represented at all stations (Figure 1). To meet the Stockdon et al. (2006) equation's deep-water wave conditions criterion, waves are linearly backshoaled to deep-water before computing wave runup.

For each study site, we define a representative coastal stretch ~30-km long with proximity to the NOAA water level station for identifying the location's beach morphology characteristics (Supplementary Tables S1 and S2). Beach morphology data (i.e., dune toe, dune crest and beach slope) are then obtained from cross-shore profiles derived from the United States Geological Survey LiDAR elevation surveys at 100-m intervals (Doran et al., 2020; Shope et al., 2021). For the US Atlantic and Gulf coasts, we use the most recent non-post-storm LiDAR survey available at each site (Doran et al., 2020), while for the Pacific coast, we use a single LiDAR survey dataset for all sites (Shope et al., 2021). To calculate wave runup, we use the mean beach slope, defined as the gradient from the dune toe to the shoreline (approximated by the Mean High Water (MHW) of the NAVD88 datum; Supplementary Text S2). We use this metric over the foreshore beach slope, as the latter varies daily with wave conditions, extreme water levels typically act over the whole beach face and storm waves often flatten the foreshore slope (Palmsten and Holman, 2012; Stockdon et al., 2012). Moreover, an experimental study showed that pre-storm mean beach slopes yielded more accurate runup predictions during dune-erosion events than foreshore slopes, while also being easier to estimate consistently at large spatial scales (Palmsten and Holman, 2012). Because the Stockdon et al. (2006) empirical runup model was parameterized using beach slopes with average values generally < 0.12, we exclude all beach profiles with beach slopes exceeding this upper limit to remain within the model's applicability range.

### Determining total water level relative composition during storm impact regimes

We use the Storm Impact Scale to assess coastal change likelihood by comparing the TWL elevation to the elevation of morphological thresholds (Sallenger, 2000; Stockdon et al., 2007). In this scale, four impact regimes are defined: swash (Mean Higher High Water [MHHW] < TWL < dune toe), collision (dune toe ≤ TWL < dune crest), overtopping/overwash (TWL ≥ dune crest) and inundation (dynamic SWL ≥ dune crest). We define the MHHW of the NAVD88 datum as the lower limit of the swash regime (instead of MHW as defined in Sallenger (2000)) because areas above MHHW are dry most of the time; thus, we assume TWLs exceeding MHHW to be driven by storms or sea level anomalies and may induce beach face changes important for beach recovery between storm events. Collision serves as a proxy for dune toe erosion, while overtopping (originally named overwash in Sallenger (2000)) is a proxy for dune erosion with landward sediment transport with flood potential, as water levels intermittently overtop the dune. The most severe regime is inundation, measured using the dynamic SWL (i.e., SWL + wave setup) instead of TWL, which serves as a flooding proxy since the beach and the dunes are continuously submerged.

To classify hourly TWLs into storm impact regimes, we first calculate wave runup at individual cross-shore beach profiles to pair with each hourly SWL in the record, which results in ~300 TWL time series at each station (i.e., 100 m-spaced profiles along a 30-km-long coastline; Figure 2a,b). The morphological thresholds at each profile are compared to hourly TWLs from 1980 to 2021 to classify the impact regime (Figure 2c). Next, we average the magnitude of the individual TWL driving processes for each regime by year (Figure 2d) and determine the percent contribution of relative sea level, astronomical tides, seasonality, sea level anomalies, storm surge and wave runup to each impact regime (Figure 2e). We exclude the datum from our relative contribution assessment by averaging the relative sea level component and subtracting this constant value from the relative sea level, thus isolating the contribution of long-term sea level trends. We repeat this process at all beach profiles, compute the average contribution of each process to each regime across profiles and compare it across stations (Figure 2f).

Because beach morphology changes over time, we test the robustness of using a single, static survey to represent beach slope and morphological thresholds at Duck, NC, and Pensacola, FL, where multiple LiDAR surveys are available. In these cases, we replicate our analysis using (i) long-term averaged beach slopes and thresholds from multiple LiDAR surveys and (ii) time-varying beach morphology, updating beach morphology values whenever a new survey was available. Average TWL relative composition during impact regimes varies only slightly when multiple surveys are considered compared to using a single static morphological survey (Supplementary Table S3 and Supplementary Figures S1 and S2). The rank order of dominant TWL contributors to each regime remains unchanged, and static-morphology contributions generally fall within the variability of those based on time-varying morphology. This demonstrates that our static approach is sufficient to characterize the dominant drivers of potential coastal impacts.

### Assessing total water level magnitude variability

To assess whether TWL magnitude varies across storm impact regimes, we perform one-way ANOVAs. First, we extract hourly TWL magnitudes during impact regimes at each study site and compute the average TWL magnitude per regime per site. We group sites into seven regions based on distinct coastal geography, guided by the US climate regions (Karl and Koss, 1984). These regions include the PNW (Washington and Oregon), California (CA), West Gulf (Texas and Louisiana), East Gulf (Florida), Northeast (New Jersey and Delaware) and the Southeast, which is further

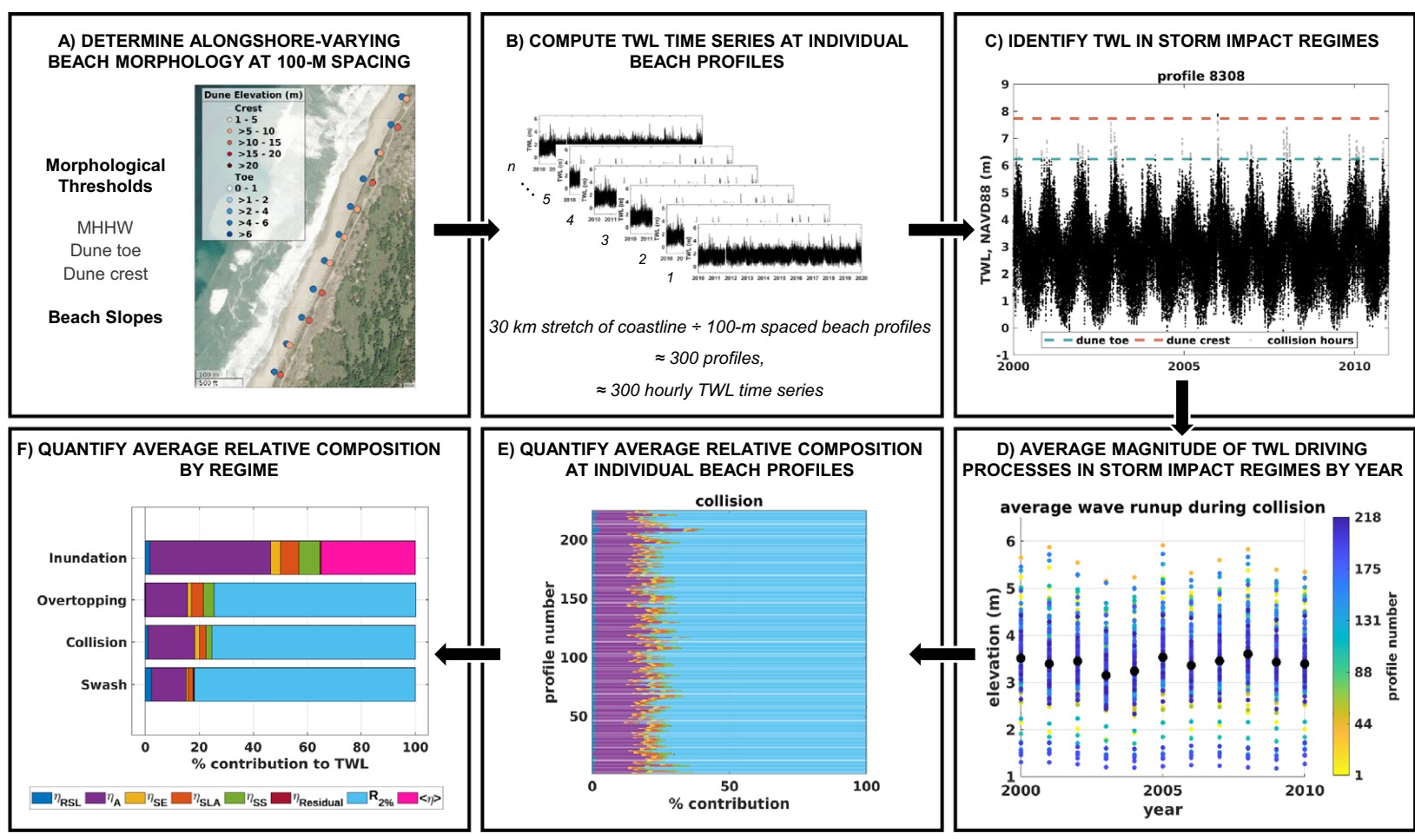

**Figure 2.** Example of the steps for determining the relative contribution of individual water level components to the total water level (TWL) during storm impact regimes in Charleston, OR, for the collision regime. Colors in the contribution plots represent processes as follows: relative sea level ($\eta_{RSL}$, dark blue), astronomical tide ($\eta_A$, purple), seasonality ($\eta_{SE}$, yellow), sea level anomalies ($\eta_{SLA}$, orange), storm surge ($\eta_{SS}$, green), residual ($\eta_{residual}$, red), wave runup ($R_{2\%}$, light blue), for swash, collision and overtopping, and wave setup ($<\eta>$, pink), for inundation.

distinguished into the northern Southeast (N-SE; Virginia and North Carolina), and the southern Southeast (S-SE; South Carolina, Georgia and Florida), reflecting the latitudinal gradient in the proportion of extreme TWLs associated with tropical and extra-tropical cyclones along the US Atlantic coast (Quadrado and Serafin, 2024). Next, we average the site-averaged TWL magnitudes within each region by regime, yielding three TWL averages per region, one per regime (i.e., swash, collision and overtopping). The inundation regime is excluded from ANOVA testing due to a considerably smaller sample size than the other regimes, as only a few hours of inundation are detected at most locations. We use these regionally averaged TWL values to conduct two sets of ANOVA tests. First, to evaluate whether average TWL magnitudes differ across regimes within a region, we perform seven ANOVAs, one for each region, using the regionally averaged TWL magnitudes for each regime (number of levels = 3). Second, to assess whether average TWL magnitudes differ across regions within a regime, we perform three ANOVAs, one for each regime, using the regional TWL averages (number of levels = 7).

### Linking water level percentiles to morphological thresholds to assess the probability of potential impacts

We evaluate the probability of storm impact regimes occurring by linking TWL percentiles to the elevation of morphological thresholds representative of each storm impact regime. Percentile ranks are assigned to TWL elevations matching each regime threshold using the empirical cumulative distribution function (ECDF) of hourly TWL data. For the inundation regime, the dynamic SWL ECDF of hourly data is used. The percentiles indicate how frequently the TWL or dynamic SWL reaches a threshold, allowing us to identify regions with higher potential for experiencing coastal impacts.

## Results

### Total water level magnitude variability

Figure 3 presents the empirical distributions of TWL, SWL and wave runup magnitudes during swash, collision and overtopping regimes across the US coastal regions. TWL magnitudes are statistically different across regimes at all regions (Supplementary Table S4). Overall, the average and standard deviation of TWL, SWL and wave runup magnitudes gradually increase from swash to overtopping (Supplementary Table S5). From swash to collision, TWL, SWL and wave runup distribution tails lengthen, and from collision to overtopping, tails become generally heavier (Figure 3). Regional differences in TWL, SWL and wave runup within each regime are also statistically significant. The PNW and CA exhibit the highest TWLs and standard deviations across regions, with averages increasing by 1.9 and 0.8 m from swash to collision and collision to overtopping, respectively, in the PNW, and by 1.4 and 1.3 m, respectively, in CA for the same transitions. In contrast, the Gulf regions show the lowest TWLs and narrowest distributions but still experience absolute increases comparable to those along the Pacific coast, with average TWLs increasing by 1.3 m from swash to collision and 1.2 m from collision to overtopping. Similarly, average SWL magnitudes are highest in the PNW and CA and lowest in the Gulf (Figure 3d–f), with the largest increases from swash to collision in the PNW and NE, and from collision to overtopping in the N-SE and Gulf regions. Across all regions, wave runup presents the largest average magnitude increases, about 1.1 m from swash to collision and 1.0 m from collision to overtopping, compared to smaller average SWL increases during the same regime transitions at 0.5 and 0.2 m, respectively. While the Pacific coast exhibits the highest average wave runup magnitudes during swash, averages in the NE and N-SE become comparable to those along the Pacific during the collision and overtopping regimes (Figure 3g–i).

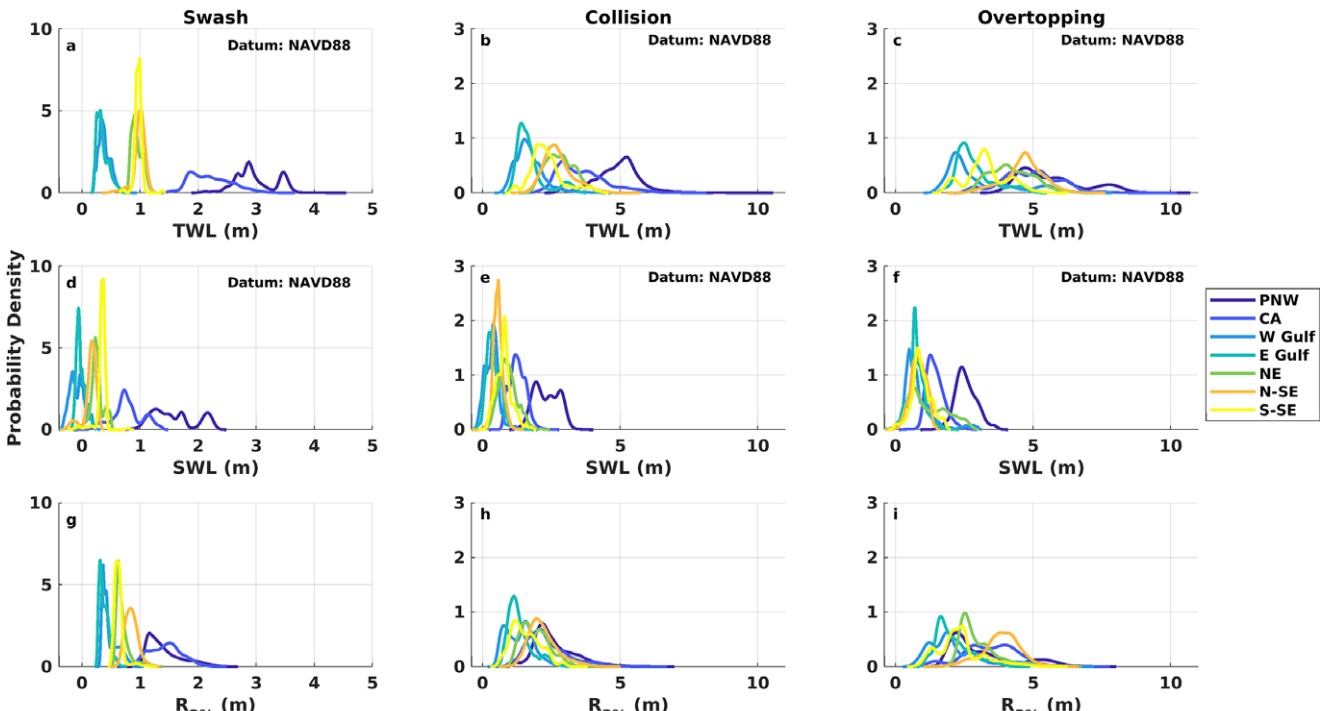

**Figure 3.** Empirical probability density functions of total water level (TWL), still water level (SWL) and wave runup ($R_{2\%}$) during the swash (a), (d), (g), collision (b), (e), (h) and overtopping (c), (f), (i) regimes. The Pacific coast is represented by the Pacific Northwest (PNW) and California (CA) regions, the Gulf coast is represented by the West Gulf (W-Gulf) and East Gulf (E-Gulf) regions and the Atlantic coast is represented by the Northeast (NE), Northern Southeast (N-SE) and Southern Southeast (S-SE) regions.

### Total water level composition during storm impact regimes

Figure 4 displays the average TWL relative composition during swash, collision, overtopping and inundation events at four representative stations along each US coastline. Statistics reported in this section are quantified using all 26 study sites unless specified. The top three largest contributors to TWLs in the swash regime are wave runup (70%), astronomical tides (20%) and relative sea level (6%), although at some sites contributions from storm surge and/or sea level anomalies exceed those from relative sea level, like Charleston, Grand Isle, Springmaid Pier and Duck (e.g., Figure 4b,f,j,k). As impact regimes intensify from swash to collision to overtopping, average storm surge contributions increase by 6% and 3%, respectively, while average tidal contributions decrease by 3% and 4% across the same transitions. On average, wave runup contributes ~67% of TWLs during collision and overtopping, although at some sites (e.g., La Jolla, Apalachicola, Lewes; Figure S1j, o, x) its contribution decreases by up to 20% compared to swash as regimes become more impactful. Inundation is, on average, dominated by waves (38%), storm surge (26%) and tides (20%). Sea level anomalies and seasonality, often the lowest contributors to TWLs in all regimes, increase in contribution as regimes intensify, with sea level anomalies' contribution peaking during inundation.

Although broad patterns in the relative contribution to TWLs across regimes exist, the composition of the TWL varies across impact regimes at both regional and local scales. For example, wave contributions are highest along the Pacific coast, ranging from 46% (inundation) to 82% (swash). In contrast, the lowest wave contributions occur along the Gulf coast, ranging from 27% (inundation) to 67% (collision). Storm surge influence is greatest along the Gulf coast, ranging from 4% (swash) to 48% (inundation), and lowest along the Pacific coast, where its contribution to impact regimes ranges from 0.3% (swash) to 10% (inundation). At several sites, storm surge contributions during inundation exceed regional averages by up to 24%, for example, at Toke Point and South Beach in the Pacific, Galveston and Apalachicola on the Gulf and from Charleston to Fernandina Beach on the Atlantic (Supplementary Figure S1). Tidal contributions are highest on the Atlantic (21%) and the Pacific (23%) coasts, and lowest on the Gulf coast (5%).

### Probability of potential impacts

Figure 5 shows empirical percentiles of TWL (for swash, collision and overtopping regimes) or dynamic SWL (for the inundation regime) at site-specific morphological thresholds, estimating the

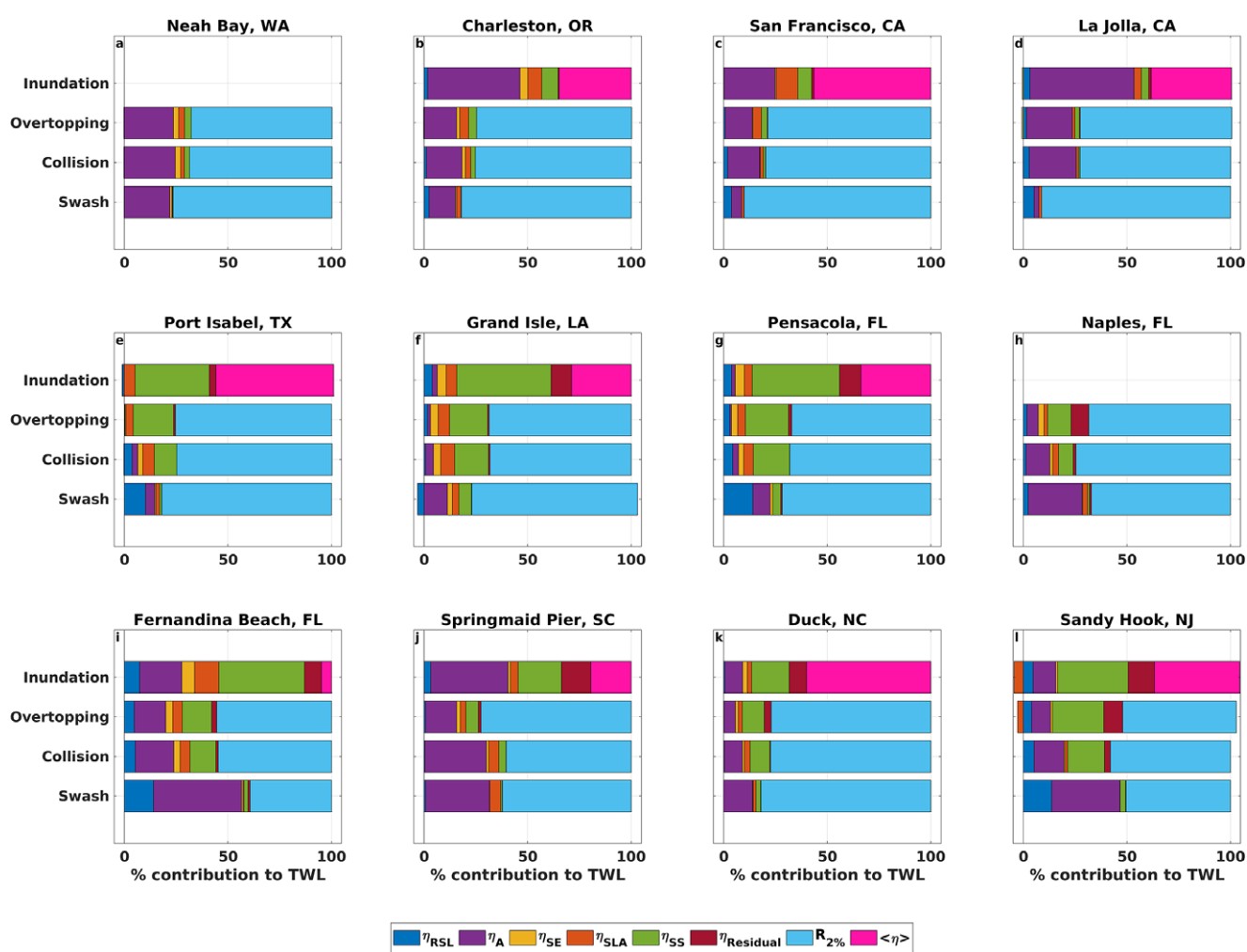

**Figure 4.** Average percent contribution of individual water level components to total water levels (TWLs) during swash, collision, overtopping and inundation storm impact regimes from 1980 to 2021 at stations along the US Pacific (a)–(d), Gulf (e)–(h) and Atlantic (i)–(l) coastlines. Colors indicate the following contributors: relative sea level ($\eta_{RSL}$) in dark blue, astronomical tide ($\eta_A$) in purple, seasonality ($\eta_{SE}$) in yellow, sea level anomalies ($\eta_{SLA}$) in orange, storm surge ($\eta_{SS}$) in green, residual ($\eta_{residual}$) in red, wave runup ($R_{2\%}$) in light blue and wave setup ($<\eta>$) in pink. Regimes with no events occurring are left blank.

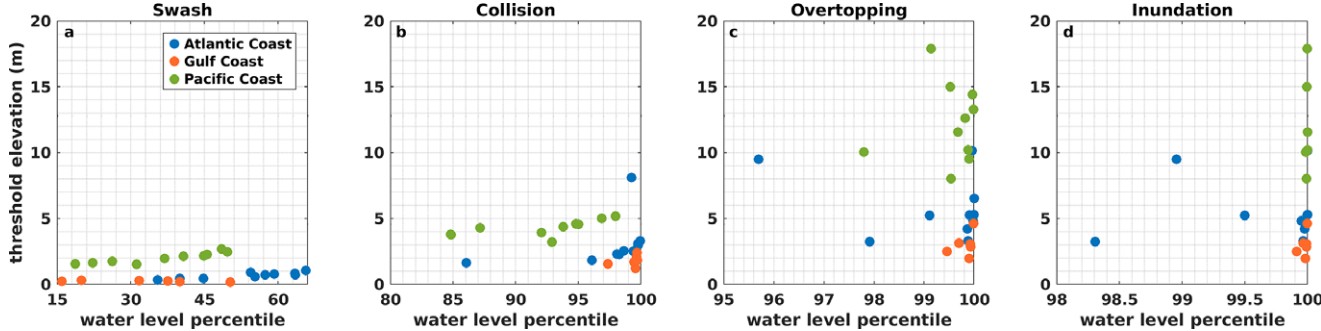

**Figure 5.** TWL or dynamic SWL percentiles of morphological threshold elevations for swash (a), collision (b), overtopping (c) and inundation (d) regimes. Elevation values are relative to NAVD88. Colors distinguish among the US coastlines: the Atlantic coast in blue, the Gulf coast in orange and the Pacific coast in green.

likelihood of those magnitudes. Lower percentiles indicate a higher likelihood. Swash is the most frequent regime at all locations, occurring on average at the 34th TWL percentile along the Pacific and Gulf coasts compared to the Atlantic coast (~54th percentile; Supplementary Table S6). Collision is less frequent than swash and is more likely to occur along the Pacific (92.0th ± 4.8 percentile) and Atlantic (97.5th ± 4.2 percentile) coasts than on the Gulf (99.2nd ± 0.9 percentile). Overtopping and inundation are the least frequent regimes, occurring on average across all US coastlines at the 99.5th ± 1.0 TWL and 99.8th ± 0.4 dynamic SWL percentiles, respectively, although the Atlantic coast is more prone to experiencing such regimes than the Gulf and the Pacific.

## Discussion

### Variability in total water level magnitude and composition across storm impact regimes

As coastal impacts intensify from the swash to the inundation regime, TWL magnitudes gradually increase and their relative composition changes. While previous studies identified waves as the main driver of extreme TWLs (i.e., annual maximum, 100-year event) along US coastlines (e.g., Serafin et al., 2017; Parker et al., 2023; Quadrado and Serafin, 2024), our results show that even more frequent events, like those in the swash regime, are also wave-dominated. Some of the highest average wave runup magnitudes and contributions to swash occur along the Pacific and N-SE Atlantic regions, where beach slopes are among the steepest across our sites (Supplementary Table S2), and wave energy is greater than in the Gulf, NE and S-SE Atlantic (Lehmann et al., 2017; Ahn et al., 2020). Tides, the second largest contributor to swash, are most influential at Pacific and Atlantic coast sites, where mean tidal ranges are largest (Supplementary Table S2). Relative sea level is another important contributor to swash, particularly along the Gulf and Atlantic coasts, where the fastest sea level rise rates generally align with the largest relative sea level contributions (Supplementary Figure S1 and Supplementary Table S2). For instance, the highest relative sea level contribution to TWLs during swash (29.5%) occurs in Galveston, TX, which has the second-highest relative sea level rise rate across stations (i.e., 6.65 mm/year; NOAA CO-OPS, 2025). This is consistent with evidence that diurnally dominated, microtidal regions like the Gulf are highly sensitive to sea level rise, as even small increases in sea levels can rapidly lengthen the periods during which water levels remain above flooding thresholds relative to tidal datums (Talke, 2025). Collectively, these results suggest that rising seas could increase the frequency of coastal impacts by raising baseline water

levels, which increases threshold exceedance likelihood (Sweet and Park, 2014; Hague et al., 2020; Taherkhani et al., 2020; Almar et al., 2021; Hames et al., 2023).

TWLs in the collision regime remain primarily wave-driven (Figure 4), as waves become more energetic during storms (e.g., Bromirski and Kossin, 2008; Marsooli and Lin, 2018). However, the waves' relative contribution is generally offset by increasing SWL magnitude and contribution as TWLs shift into higher-impact regimes (Figures 3 and 4). Along the Atlantic and Gulf coasts, increasing SWL contributions from swash to collision are mainly attributed to storm surge, which increases by 8% and 12%, respectively. In contrast, on the Pacific coast, where storm surge contributes only 2% on average to collision, the increase in SWL contribution from swash to collision (8%) is driven mainly by tides. Differences in the proportion of waves and SWLs influencing TWLs can lead to distinct coastal responses during collision. For example, field observations and simulations in the PNW show that wave-driven dune toe accretion can occur on low-sloping beaches when the dynamic SWL remains below the dune toe during collision (Cohn et al., 2019). Variations in TWL relative composition also introduce uncertainty in coastal hazard projections. On the US Pacific coast, a probabilistic hazard assessment reveals that wave runup and astronomical tides are the main contributors to collision (Leung et al., 2024). Such variability in wave and SWL magnitude adds uncertainty to collision and overtopping projections that can exceed uncertainty associated with sea level rise through the twenty-first century, since small TWL changes around regime thresholds can determine impact occurrence (Leung et al., 2024).

Sea level anomalies and seasonality also increase in relative contribution from swash to collision, further offsetting wave runup contribution despite typically being the smallest contributors to impact regimes. Sea level anomalies can exacerbate coastal hazards, as they are modulated by large-scale climate patterns such as the Atlantic Multidecadal Oscillation on the US Atlantic and Gulf coasts (Park et al., 2010; Park and Dusek, 2013) and the El Niño Southern Oscillation along the US Atlantic and Pacific coastlines, which has been linked to coastal erosion through its influence on sea level anomalies and wave forcing (Park and Dusek, 2013; Barnard et al., 2015, 2017; Vos et al., 2023). Observations from a US Atlantic barrier island over 4 years showed higher erosion rates during a year with positive sea level anomalies and no intense storms compared to a year impacted by a major hurricane (Theuerkauf et al., 2014). Moreover, seasonal flooding on the US Atlantic coast correlates with seasonal sea level variability, such as annual and semi-annual tidal cycles and seasonal Gulf Stream flow changes (Ezer, 2020), while in the PNW, winter storms coincide

with downwelling and warmer waters that increase coastal water levels and flood potential (Kniskern et al., 2011; Durski et al., 2015). These examples emphasize the relevance of smaller TWL contributors to coastal impacts.

During overtopping, wave runup remains the dominant TWL contributor, with contributions similar to those during collision, while storm surge contributions further increase across all coastlines. Increases in storm surge magnitude are reflected in longer, heavier TWL and SWL distribution tails during collision and overtopping compared to swash, suggesting a growing influence of high-storm surge events as regimes intensify. Storm surge influence on coastal flooding and erosion has been well-documented (Vellinga, 1982; Houser et al., 2008; Gharagozlou et al., 2020; van Wiechen et al., 2023), with contributions to SWLs reaching twice those of annual maxima during less frequent events, such as the 100-year TWL (Serafin et al., 2017). Elevated storm surge magnitudes and prolonged storm durations during events such as Superstorm Sandy in 2012, which produced widespread storm surge exceeding 1 m along the US Atlantic coast (Blake et al., 2012; Sopkin et al., 2014; Schubert et al., 2015), have been linked to increased overtopping likelihood since SWLs have more time and energy to erode frontal dunes, increasing a beach's vulnerability to impacts (Donnelly et al., 2006). Storm surge contribution peaks during the inundation regime, but at some locations, this reflects only a few storm events, like Hurricanes Isaac (2012) and Ida (2021) at Grand Isle, LA, where storm surge exceeded wave setup contribution (Figure 4). Given the limited number of inundation hours, TWL composition during inundation may be similar to that of overtopping at the same site, especially considering that overtopping may lower dune elevations and inundate dune systems as storms progress (Stockdon et al., 2007; Long et al., 2014).

### Probability of US coastlines experiencing impacts

Although impact regimes are defined consistently by morphological threshold exceedance, regional variability in climatology and coastal morphology affects coastal impact likelihood. Study sites across the Atlantic and the Pacific coasts are more likely to experience impacts than the Gulf (Figure 5 and Supplementary Table S6), despite the Gulf and Atlantic having similar morphological thresholds and the Pacific having the highest regime thresholds (Supplementary Table S2). This reflects more energetic wave climates (Lehmann et al., 2017; Ahn et al., 2020) and higher mean tidal ranges (Supplementary Table S2) along the Atlantic and Pacific, which increase the probability of threshold exceedance. For instance, TWL simulations during Hurricanes Matthew and Dorian show that more locations along the US Atlantic coast experienced dune toe and crest exceedances when peak TWLs were considered, rather than the combination of peak storm surge and wave runup alone, demonstrating the importance of tides to impact occurrence (Hsu et al., 2023). On the Pacific coast, beaches near the California-Oregon border and Northern Washington are more susceptible to collision and overtopping than adjacent regions due to their steep beach slopes (> 0.1) and relatively low backshore feature toes (Leung et al., 2024). In contrast, the Gulf coast exhibits storm surge contributions averaging two to eight times greater than those along the Atlantic and Pacific, yet remains the least likely to experience collision, overtopping and inundation impacts because high storm surges are generally less frequent than large waves.

### Limitations and assumptions

Because long-term consistent surveys of beach morphology are lacking, we use static beach morphology datasets representative of one snapshot in time to represent dune and beach morphology over a 41-year period. However, beach morphology is highly dynamic; thus, the beach slope and morphological thresholds used herein may not represent conditions throughout the full study period, potentially over- or underestimating impact regime occurrence due to its influence on wave runup magnitude and storm impact regime frequency. We tested the robustness of our TWL relative contribution results by comparing our approach of choosing one representative LiDAR survey to define morphology to the average slope of multiple LiDAR surveys and a time-varying approach, which replaces morphological parameters over the course of the time series at Duck, NC, and Pensacola, FL, similar to the methods applied in Serafin et al. (2019), who demonstrated that accounting for temporal variability in beach slopes is particularly important for extreme TWL magnitude estimation on profiles with high beach slope variability. Our results show that average TWL composition during impact regimes remains consistent when using a single or multiple morphology surveys (Supplementary Text S3 and Supplementary Figures S2 and S3). Similarly, Kim et al. (2023) demonstrate that wave runup estimates during storms derived from constant, pre-storm beach slopes have similar accuracy to those derived with time-varying slopes, supporting our approach. While high-temporal-resolution (e.g., daily, weekly, monthly) datasets are lacking nationally, our use of spatially varying beach morphology captures local variability, reducing potential uncertainties from spatially averaging morphology. Moreover, local US monitoring programs provide event-based to seasonal data (e.g., Gibbs et al., 2001; Allan and Hart, 2008; Doran et al., 2020), but the lack of consistent temporal resolution across our study sites would limit our analysis to a few locations if we choose to only do our analysis on locations with a comprehensive long-term morphology dataset.

We use the shallowest depth contour available across US coastlines (i.e., ~100 m isobath), but this does not fully capture regional bathymetric differences, as the 100 m contour is at different distances from the coast (Figure 1). Thus, wave transformation occurring between the 100 m contour and the nearshore is not included, and significant wave heights used to estimate wave runup may be higher than those at the coast. Extreme TWLs estimated from nearshore-transformed waves that were backshoaled to deep-water differed in the PNW by 20–40 cm in magnitude from those derived from shelf edge, deep-water waves (Serafin et al., 2019), yet > 85% of extreme TWLs are driven by the same offshore wave conditions when accounting for shoaling effects (Serafin et al., 2019). Despite partial inclusion of wave transformation, the wave datasets used here effectively represent dominant extreme wave characteristics driving coastal change on sandy shorelines.

### Conclusions

In this study, we show that TWL magnitude and composition vary across regimes and regions, influenced by local climatology and morphology, ultimately affecting the probability of potential coastal impacts along US coastlines. Wave runup is the main driver of TWLs across impact regimes, but its contribution is gradually offset from swash to inundation by increasing contributions from storm surge on the Atlantic and Gulf coasts and by tides along the Pacific, despite waves becoming more energetic during storms. Increasing contributions from sea level anomalies and seasonality across regimes also offset wave influence, demonstrating that even minor TWL contributors can amplify coastal change potential. Storm surge influence progressively increases from swash to inundation

as regimes intensify, peaking during overtopping and inundation, even along the Pacific coast, where its influence is typically low. The contribution of other SWL processes to regimes, such as relative sea level and tides, also depends on location. The largest relative sea level contributions are observed along the Gulf coast, where sea level rise rates are highest, while larger mean tidal ranges along the southeast Atlantic and PNW coasts result in higher tidal contributions.

While impact regimes are consistently defined by morphological threshold exceedance, the TWL magnitude and composition driving regimes are location-specific, influencing where and how often impacts occur. The Atlantic and Pacific coasts are more susceptible to storm impact regimes than the Gulf coast due to relatively higher TWLs, driven by more energetic wave climates and larger mean tidal ranges compared to the Gulf. On the Gulf coast, storm surge contributions to impacts can be up to 20 times greater than along the Atlantic and Pacific coasts, but impacts occur less frequently than along these coastlines because high storm surge magnitudes are generally less frequent than large waves.

Our work contributes to understanding spatial variations in coastal impact drivers along the sandy coastlines of the United States, which is essential considering projected changes in sea levels, wave climate and storminess, and the growing demand for projecting and adapting to current and future coastal changes at the local scale.

**Open peer review.** To view the open peer review materials for this article, please visit http://doi.org/10.1017/cft.2025.10018.

**Supplementary material.** The supplementary material for this article can be found at http://doi.org/10.1017/cft.2025.10018.

**Data availability statement.** Beach morphology data (i.e., dune toe elevation, dune crest elevation and beach slope) are derived from publicly available LiDAR-based surveys conducted by the United States Geological Survey (USGS) (Doran et al., 2020; Shope et al., 2021). Details of the specific surveys and locations used in this study are provided in Supplementary Table S1 in the Supporting Information. Raw hourly still water level records are obtained from the National Oceanic and Atmospheric Administration (NOAA) National Ocean Service (NOS) Tides and Currents database (https://tidesandcurrents.noaa.gov/). The locations of the water level stations used in this study are listed in Supplementary Table S2 of the Supporting Information, and their identification number is shown on Figure 1 in the main text. Wave climate data, including significant wave height and peak wave period, are sourced from the Global Ocean Waves (GOW) version 2.0 hindcast database (Pérez et al., 2017) developed by the Institute of Environmental Hydraulics of the Universidad de Cantabria (IHCantabria). These wave datasets can be requested freely for academic use via IHCantabria's contact portal (https://ihdataprot-a.ihcantabria.com/contact-us/). A data repository including the main analysis script used to reproduce our workflow and example data from one representative station in our study is available on Zenodo at https://zenodo.org/records/17418447.

**Acknowledgments.** The authors would like to thank Li Erikson, at the United States Geological Survey (USGS Pacific Coastal and Marine Science Center, and Kara Doran, at the USGS St. Petersburg Coastal Marine and Science Center, for assisting with obtaining consistent LiDAR-derived beach morphology surveys across the United States coastlines. The authors would also like to thank Melisa Menéndez at the Environmental Hydraulics Institute of the Universidad de Cantabria (IHCantabria) for providing the Global Ocean Wave version 2 (GOW2.0) data and Meagan Wengrove, whose questions contributed to the presentation of the empirical probability distributions. The authors would also like to thank three anonymous reviewers whose comments improved this manuscript.

**Financial support.** Gabrielle P. Quadrado is supported by the University of Florida Graduate School Fellowship.

**Competing interests.** The authors declare none.

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
