## [Reviewer Report]

Overview

This study evaluates the spatial variability, magnitude, and relative composition of total water level at various locations along US coastlines. The authors classified the study regions into 4 regimes and quantified the probability of each regime to experience impacts. The authors investigate the importance of different components and aspects of the Sallenger regimes contained in the total water level.

The study objective and the adopted methodology are mostly clear. The authors presented their results in a clear way, carried out a well-structured discussion on their findings, and contextualized their findings with some existing research. The manuscript is well-written and well-structured.

I recommend publication after minor revision

Major comments:

1) This study can benefit from some additional comparisons with more recent previous studies addressing TWL and various components to flooding.

2) The study is conducted regionally and I can understand why. However, some discussion on variability within regions (not captured) from the present approach is needed. For example beach slope, dune variability etc. Also am I correct that no station were used on the east of Florida south of Fernandina? Why?

3) The study could benefit from a sensitivity analysis.

Example: Only a single lidar survey could be used that would not represent the mean or possibly even typical conditions of a dynamic area.

Example: the beach slope used in the Stockdon formula seems problematic. The authors use the gradient from the dune toe to the MHW. That slope will certainly be different than the true foreshore slope. The back beach is often “flat” or may even have a negative gradient if there is a pronounced berm. Why not choose a definition more commensurate with the foreshore slope… like +/- 1 m either side of mean water, or +/- 2 m.

Example; Do all beaches used have dunes? La Jolla is mostly backed by cliffs for example. What is done in these scenarios and does the Stockdon formula make sense with how the slope is being defined?

4) The writing throughout is passive.

Minor comments:

1) Line 86: “Gulf of Mexico” “Gulf of America” now. Also fix Figure 1

2) Line 136: TWL and SWL were defined before, use the abbreviations instead.

3) Line 138: This equation is so basic that it is not needed

4) Line 155: The Stockdon formula is used for runup but it is far from “perfect”. Some studies have shown the R2% from this approach underestimate erosion (Palmsten and Holman 2012) AND this type of simple formula lacks explicit treatment of the dune and foreshore geometry (Park and Cox, 2016) beyond the foreshore slope.

5) Line 202: “To identify whether an hourly TWL is in a specific regime”. This phrase is not clear.

6) Figure 2: the legend and the text are not easily readable in subplots (1) and (5).

7) Lines 249, 251: define n. I supposed the number of ANOVA tests (7 then 3). Or degrees of freedom?

8) Figure 3: The y-scale is different between subplots. Suggesting unifying the y-scale for each variable for easier comparison.

9) Figure 4: The legend is too small. The caption is too long.

10) Line 359: I imagine this statement comes from the particular locations where data were extracted. However, blanket statements on slopes being “steeper” for the broad regions is misleading.

---

## [Reviewer Report]

The manuscript by Quadrado & Serafin examines the importance of considering not only total water level (TWL) magnitude but also its composition when determining coastal impacts using Sallenger’s storm impact scale. The authors emphasise that understanding spatial variations in TWL composition is important for prioritising which factors to consider in preparing for coastal hazards and, consequently, for improving coastal adaptation planning. Furthermore, they compare TWL magnitudes within each impact regime across multiple locations, assessing the probability of each site experiencing each regime.

I find the article to be of high quality, well structured, clearly written, and supported by good and valuable results presented in detail. The authors acknowledge the limitations of their approach in a clear manner. The study yields interesting findings regarding TWL composition and local variability, of relevance to both the coastal research community and coastal managers. Overall, my impression is very positive, and I recommend it for publication. Nonetheless, I do have some suggestions that may help to improve the manuscript.

Specific comments:

Line 102: You use the term overtopping instead of overwash. Most studies retain Sallenger’s original term. Is there a reason to use overtopping instead of overwash? While I don’t see a problem with your choice, please clarify in the text that this is your terminology change and that Sallenger originally used overwash.

Line 143: Not all NOAA station data are complete (minimum 75% hourly data over the record). Please specify how missing data were handled. Were they interpolated? Please clarify this in the text to improve methodological transparency.

Line 198: Consider whether the same term “flooding” should apply to both overtopping/overwash and inundation. For the first, overtopping may be more precise. If both are described as flooding, the reader might be confused. Changing the term flooding to inundation for the second could be a possibility (although I’m not sure if it is the correct term in this case). But then a definition distinguishing flooding from inundation would be needed.

Figure 2: Looking at the methodology of figure 2, would it be possible to include an analysis of the duration each regime dominates annually? I’m curious about it. Especially how many hours per year the inundation regime occurs, and whether this duration changes over time for each location.

Figure 3: There seem to be negative SWL values, why are these not included in the plots? Or are these data being excluded? Also, since axis ranges are very similar (e.g., 8 and 10), using the same axis limits across panels (same x for all, same y for each column) would greatly improve comparability.

Line 279: Please clarify whether values refer to PNW or CA, or if they are a median across both. If the latter, I suggest presenting median TWL values for each region separately. For the Gulf regions, this distinction seems less critical given their similarity.

Line 299: Please clarify what you mean by “tides.” Does this refer exclusively to astronomical tides? If so, by looking at the figure (although I understand the statistics represent all the stations), it seems surprising to have more RSL than SS (since you mention 4% for RSL, I assume SS is less than this). Please clarify this in text.

Line 300: Could you specify in the text which exceptions you refer to in Figure 4b–c? They are not visually apparent to me.

Figure 4: What do the values below 0 and above 100 mean in the plots? Values below 0 presumably represent negative contributions, but how does the greatest relative sea level trend at Grand Isle produce a negative contribution to TWL? Similarly, clarify the meaning of values above 100. These points are currently unclear.

Figure 5: I really like this figure. In line with my earlier comment on Figure 2, could you also present regime occurrence in hours/year? For example, on average, how many hours per year does each location experience inundation? While you mention something in line 371, a dedicated figure showing temporal evolution would be more informative and could also help visualise the increasing RSL impact (as discussed from line 374 onward).

Minor corrections:

Line 78: The statement on the lack of real-time measurements linking coastal change with hourly TWLs is somewhat inconsistent with your own approach, where you combine different datasets (wave hindcast, measured water levels, LiDAR) to do exactly that. Using separate datasets is not a problem in itself, especially since you address this in the limitations section. The issue may be the use of the term “datasets.” Consider rephrasing to “separate models (or simulations) for water levels and beach morphology” to clarify the intended meaning.

Methods: The water level datum is not specified, though I assume it is NAVD88 (same as for beach slope). Please confirm and state this explicitly. Also, the length of each dataset is not stated. Only on line 206 you defined the temporal scale for TWL. This information should be given earlier in the text.

Line 205: Add letters in the text and on each panel of figure 2 to clearly reference locations in the figure, or refer to the numbering in figure 2 when describing locations in the text (line 205 and subsequent sentences).

Line 332: Consider rewriting as: “Figure 5 displays the empirical percentiles of TWL (for swash, collision, and overtopping regimes) or dynamic SWL (for the inundation regime) corresponding to…” Also, please review and adjust the figure 5 legend accordingly for precision.

Finally, I would strongly encourage you to make your code freely and readily available, and not only “upon request”.

---

## [Reviewer Report]

Summary:

This manuscript quantifies differences in the drivers of total water levels (TWLs) across US sandy beaches. Wave hindcast and water level data from 26 locations are used to calculate hourly TWL time series. TWLs are compared to LiDAR-extracted morphology to determine the amount of time spent in Sallenger Storm Impact Regimes.

TWL composition varies spatially, temporally, and across Sallenger Regimes. Wave runup is found to be the main driver of TWLs in swash, collision, and overtopping regimes across all regions. In overtopping and (rare) inundation regimes, storm surge and tidal contribution grows – with storm surge playing a larger role in the Atlantic and Gulf Coasts, compared to Tides in the Pacific. Atlantic and Pacific coasts experience more frequent coastal impacts than the Gulf Coast due to their more energetic wave climates, despite higher morphological thresholds in the Pacific. The Gulf Coast experiences rarer but more severe storm surge contributions due to hurricane events.

Overall evaluation:

This manuscript addresses a valuable contribution to coastal hazard literature. Similar analyses have been conducted on a regional-scale but this is the first assessment that is conducted for the entire continental US.

The paper is well-written, methods and results are clearly presented. I find this paper makes a meaningful contribution to the literature, but could be strengthened, particularly in the discussion. I outline these points in the comments below.

Major comments:

1. Line 176-186: Please reference Table S3 in this section to help the reader understand the different morphologic setting of each site. If there is room, the authors could put this information in the main paper rather than hidden in the supplemental as it directly informs the TWL calculations and storm impact regime results.

2. Related to the comment above, the Stockdon run up formula is designed for gently sloping to intermediate beaches (< 0.12) backed by sandy dunes. How does this analysis deal with transects with steeper beach slopes? Did you also ensure all the relevant transects are actually backed by dunes rather than cliffs, riprap, infrastructure, other beach types that would not be appropriate to use the Stockdon formula on?

3. Line 371: It would be interesting to have a deeper exploration of the non-stationarity within the observed record. How are figures 4 & 5 different if you break up the record in different time periods? Alternatively, could authors create a figure that highlights comparisons with the past that you allude to (e.g., show that 67% of the swash hours occur between 2000-2022 for Grand Isle. How does this value compare to other locations?).

4. Line 376: Authors state that results indicate that coastal hazards will increase with SLR. Can they be more explicit about what their analysis of TWL contributions suggests for future impacts across different regions and regimes? How will the relative contributions of TWLs change in different regions & regimes in the future due to SLR? Are there trends in the observed data?

5. Discussion: I think the manuscript could benefit from having a paragraph referring back to the previous literature that explored relative contributions of TWLs in the US / globally that they described in the introduction (lines 61-77, lines 111-120). Is this work in line with previous studies? How does this study expand from previous studies?

Minor comments:

Line 89-93 : “…models are typically applied at specific locations and times as they become computationally expensive for spatial scales of thousands of kilometers... integrate detailed model insights across broader spatial [and temporal] scales” Add a description for timescales that are generally computationally expensive.

Line 96: Leung et al., 2024 (https://doi.org/10.1029/2024EF005523) is another recent example of a hazard proxy analysis, which may be relevant for your discussion of impacts in the PNW.

Figure 2: in box 6 it appears some text on your x axis was cut off.

Figure 2: in box 6- you describe the colors in your figure caption, but why not match the colors in box 5 and box 6, and the visual legend in box 5 to maintain consistency?

Figure 2: box 4 has two contrasting titles describing the ‘average magnitude of TWL’ versus the ‘average wave runup’ – you describe this in the caption, but – Why? Why not just show the TWL?

Figure 2 : You could also add the dune toe and dune crest elevations to the plot in box 4, similar to in box 3.

Line 234 - : should write your methods / analysis in past tense: “we performed… we extracted… and computed… Sites were then grouped… ect.”

Line 244-246 “Next, we average the site-averaged TWL magnitudes within each region by regime, resulting in three regional TWL averages per region, one per regime.” ---- I see in line 251 you explain that you aren’t including inundation. I would move this to follow the pasted statement. As the reader I was confused for half the paragraph why you weren’t assessing four regimes.

Line 256: should write your methods / analysis in past tense: “we evaluated”

Line 260: “The percentiles indicate how frequently the TWL or dynamic SWL is to reach a threshold, allowing us to identify regions with higher potential for experiencing coastal impacts” replace “is to reach” with “reaches”

Line 268: “Overall, the median and standard deviation of TWL, SWL, and wave runup magnitudes gradually increase from swash to overtopping.” Are these values reported somewhere?

Figure 4: please make the legend (and it’s text) larger.

Line 380: “…SWL relative contribution begins to offset that of wave runup…” Please be explicit about what regimes (or time periods?) you are comparing. Maybe could phrase as “Like the swash regime, TWLs in the collision regime are primarily driven by waves” (If I am interpreting this sentence correctly)

Recommendation:

This paper makes a valuable contribution to coastal hazards literature by expanding analysis of the relative contributions of TWL components in driving coastal hazards across the continental US. I recommend major revisions (1) to clarify the methodology/ assumptions surrounding TWL calculation for the different morphologies and (2) strengthening the discussion by expanding analysis of non-stationarity in the observed record and implications for future impacts, as well as connecting back to existing literature. I encourage resubmission once these issues are addressed.

---

## [Editor Report]

In addition to the reviewers' comments, I would emphasise the need to reconsider your approach to beach slope when calculating wave runup as this is a critical parameter that makes a large difference to the estimations. Further, I would strongly encourage you to consider including a discussion of the limitations of your work in the main body of the manuscript rather than including it as a supplmement

---

## [Reviewer Report]

The authors have done an incredibly thorough effort addressing my comments. My opinion is that the paper is ready for publication

---

## [Reviewer Report]

The authors have carefully considered and responded to all comments and suggestions from the three reviewers, incorporating all feasible revisions into the manuscript. In my opinion, the article is now suitable for publication as it is. However, as a suggestion (and only if the editor considers it appropriate), the information provided in response 1.3c could also be included in the Supplementary Information, specifically regarding the case of La Jolla, CA, where seawalls and ripraps are present, and how these coastal protection structures were accounted for in the analysis. I believe this additional detail could be useful not only for readers familiar with the region but also for those interested in methodological approaches for sites where such structures influence coastal processes.

---

## [Reviewer Report]

I believe the authors have addressed my concerns. I appreciate their efforts to clarify their methods surrounding morphology selection and the use of the Stockdon formula. The addition of the morphology case studies improves the interpretability of their results. I thank the authors for sharing their preprint for their second article and addressing my comments regarding additions to the discussion section.